# Endoplasmic Reticulum-Targeting Two-Photon Fluorescent Probe for CYP1A Activity and Its Imaging Application in Endoplasmic Reticulum Stress

**DOI:** 10.3390/molecules28083472

**Published:** 2023-04-14

**Authors:** Chao Shi, Yan Wang, Xiangge Tian, Xia Lv, Yue An, Jing Ning, Xiulan Xin, Li Dai, Xiaochi Ma, Lei Feng

**Affiliations:** 1Second Affiliated Hospital, Dalian Medical University, Dalian 116023, China; 2College of Pharmacy, Dalian Medical University, Dalian 116044, China; 3College of Bioengineering, Beijing Polytechnic, Beijing 100029, China; 4School of Chemistry and Chemical Engineering, Henan Normal University, Xinxiang 453007, China; 5Key Laboratory of Emergency and Trauma of Ministry of Education, Hainan Medical University, Haikou 571199, China

**Keywords:** cytochrome P450 1A, fluorescent probe, endoplasmic reticulum-targeting, enzymatically active probe, two-photon

## Abstract

Cytochrome P450 1A is one of the vital subfamilies of heme-containing cytochrome P450 enzymes belonging to an important exogenous metabolizing CYP in human. The abnormal of endoplasmic reticulum (ER) may directly affect the functional activity of ER-located CYP1A and be associated with the occurrence and development of various diseases. In the present study, we constructed a selective two-photon fluorescent probe **ERNM** for rapid and visual detection of endogenous CYP1A that was localized in the ER. **ERNM** could target the ER and detect the enzymatically active CYP1A in living cells and tissues. The monitoring ability of **ERNM** for the fluctuations in functionality level of CYP1A was confirmed using ER stressed A549 cell. Based on the ER-targeting two-photon probe for CYP1A, the close association of ER state and the functional activity of ER-locating CYP1A was confirmed, which would promote the deep understanding of the biofunction of CYP1A in various ER-related diseases.

## 1. Introduction

Cytochrome P450 (CYP) is a large family of hemoproteins, which mediates various oxidative transformations in nature [1,2]. CYP1A is one of the subfamilies which consists of the structurally and functionally related isoforms CYP1A1 and CYP1A2, belonging to the important exogenous metabolizing CYP in human [3,4,5]. Imipramine, theophylline, caffeine and propranolol, and environmental toxicants including polycyclic aromatic hydrocarbons (PAH) and heterocyclic aromatic amines/amides (HAA) are all small-molecule ligands of CYP1A. Aside from mediating the metabolism of many commonly used clinical drugs, CYP1A also plays an important role in the bioactivation of pro-carcinogens PAH and HAA to the carcinogenic reactive intermediates which can bind to DNA or proteins, thereby contributing to the tumorigenesis and development of cancer and implications in some toxicity events [3]. Notably, except for the key role in oxidative metabolism of exogenous substance, the accumulating evidence supporting that CYP1A is involved in various significant biological processes, for example, the balance of oxidative state and oxidative stress, thereby were associated with a variety of diseases including liver injury, renal hypertension, hypertoxic lung injury, reproductive toxicity, etc. [6,7]. Therefore, the expression and functional activity of CYP1A in different tissues and specific organelles in cells may have some important implications for prevention and treatment of the involved diseases.

Endoplasmic reticulum (ER) is the largest organelle in cells made of a single membrane, which plays a significant role in different biosystems and mediates the various essential biological processes in cells, such as synthesis of extracellular lipoprotein, modification of proteins, and regulation of intracellular free calcium concentration [8,9]. Therefore, the change in physiological status for ER and the balance of some substances in ER are important affecting factors of the pathogenesis of many diseases [8,9]. The abnormal of endoplasmic reticulum and imbalance of the key substance may directly affect the functional activity of ER-located CYP1A and be associated with the occurrence and development of diseases [10,11]. Therefore, some relevant tool molecules are needed to study the changes in CYP1A activity in the endoplasmic reticulum under different physiological and pathological conditions, and to deeply uncover the relationship between the CYP1A activity and ER state, as well as the CYP1A activity and pathogenesis of diseases.

Fluorescence imaging technique displays many advantages that are not limited exclusively to its outstanding sensitivity, fast response, and superior spatiotemporal resolution. For this reason, this technique is nowadays widely used in environmental monitoring, drug screening, and physiological and pathological process evaluation, as well as other major areas [12,13,14,15,16,17]. Particularly, the two-photon fluorescence imaging possesses the higher resolution, deep penetration, low background, etc. that has received wide attention in recent years [18,19,20,21]. Considering the important biological functions of CYP1A, several reactive small molecule fluorescent probes that target CYP1A have been developed [22,23,24]. The CYP1A activity could be used to realize the real-time monitoring of CYP1A activity in cells and tissues, making it more accessible and facilitating further investigations of CYP1A-associated physiological and pathological processes. However, the ER-targeted two-photon fluorescent probes that could detect CYP1A activity are still lacking. Therefore, the two-photon molecular tool that can realize ER-targeted CYP1A activity imaging deserves to be explored for understanding the interaction effect between CYP1A and ER.

In the present study, we developed a two-photon fluorescent probe that could realize the activity detection of ER-located CYP1A. The fluorescent probe **ERNM** preferentially locates in ER of the cells, which attributes to the introduction of *p*-methylbenzene sulfonamide group to the distal end of the metabolic recognition site on the fluorophore naphthalimide. It was successfully used in detecting CYP1A in living cells and tissues by two-photon fluorescence imaging, thereby facilitating the location detection of CYP1A activity in complex biological systems. Using this fluorescent molecular tool **ERNM**, the activity changes in CYP1A under ER stress were effectively detected, which indicated the practical applications in uncovering the important roles of CYP1A in ER-related diseases.

## 2. Results and Discussion

### 2.1. Design and Synthesis of ERNM

Naphthalimide, as a classical fluorophore, has high fluorescence quantum yield, and the imide bond has strong electron absorption ability. Its fluorescence spectrum can be adjusted by changing the substituent group at C4 or C5, and its structural modification is relatively simple. In addition, naphthalimide has a two-photon fluorescence property, which gives it a unique advantage in biological applications. Here, *p*-methylbenzene sulfonamide group is first introduced into naphthalimide fluorophore, which has an endoplasmic reticulum targeting function, and methoxy is then introduced at the C4 site as the recognition site of CYP1A. **ERNM** was synthesized from 4-bromo-1,8-naphthalic anhydride in a two-part reaction. After the metabolism of **ERNM** through CYP1A, the methoxyl group became an oxygen anion, and the electron donor ability became stronger, resulting in significant changes in fluorescence intensity; therefore, **ERNM** could be used as a fluorescent probe for the detection of CYP1A. 

### 2.2. Spectroscopic Response of ERNM toward CYP1A

In the first instance, the spectral properties of the designed probe **ERNM** toward CYP1A were investigated. The addition of increasing concentrations of CYP1A1 and CYP1A2 led the fluorescence intensity to gradually increase (Figure 1A,C). Furthermore, the fluorescence intensity at 558 nm increased linearly by adding increasing concentrations of CYP1A1 and CYP1A2 (0 to 20 nM), and the linear equations are Y = 1371X + 523.2 (R^2^ = 0.997) for CYP1A1 and Y = 1538X + 252.1 (R^2^ = 0.991) for CYP1A2, respectively (Y means fluorescence intensity and X means enzyme concentrations of CYP1A1 and CYP1A2). Additionally, the fluorescence intensity at 558 nm gradually increased from 0 to 30 min when ERNM was incubated with CYP1A1 and CYP1A2, respectively (Appendix A). The distinct fluorescence response indicated that CYP1A-mediated demethylation of **ERNM** liberates the oxygen atom as a strong electron donor in the D−π−A structure, confirming the presence of intramolecular charge transfer (ICT) effects (Figure 1). Additionally, the demethylated product of **ERNM** exhibited a high fluorescence output signal over the pH range of 4–12, while **ERNM** consistently exhibited extremely weak fluorescence. These results indicated that **ERNM** could be used as a molecular tool to detect the activity of CYP1A in physiological process (Appendix A).

### 2.3. Selectivity Analysis of ERNM

The selectivity of **ERNM** for various CYP isoenzymes and potential interfering substances was further confirmed. The recombinant human CYP in vitro incubation assay indicated that CYP1A1/1A2 mediated the generation of fluorescence signals at 558 nm over other CYP isoenzymes including CYP1B1, CYP2A6, CYP2A13, CYP2B6, CYP2C8, CYP2C9, CYP2C18, CYP2C19, CYP2D6, CYP2E1, CYP2J2, CYP3A4, CYP3A5, CYP4A11, CYP4F2, CYP4F3, and CYP4F12 (Figure 2). Furthermore, the specificity of **ERNM** in complex biological samples was confirmed using subtype-selective inhibitors of CYP isoenzymes. The generation of the fluorescence signal can be sufficiently suppressed by CYP1A selective inhibitor (*α*-naphthoflavone), in comparison with the control group and another inhibitory group using TEPA (N,N′,N″-triethylene thiophosphoramide), montelukast, sulfaphenazolum, omeprazole, quinidine, chlormethiazole, and ketoconazole (Appendix A). Additionally, the fluorescence response of **ERNM** was hardly affected during incubation with various potential interfering species, including common ion (Mn^2+^, Ca^2+^, Mg^2+^, Ni^2+^, Zn^2+^, Sn^4+^, K^+^, Fe^3+^, Na^+^, Ba^2+^) and amino acid (Ser, Trp, Gly, Arg, Cys, GSH) (Appendix A). The relative results indicated the high specificity of **ERNM** for CYP1A in complex bio-samples.

### 2.4. Kinetic Behavior Analysis

The kinetic plot of CYP1A1/1A2 catalyzed **ERNM**
*O*-demethylation was depicted to explore the interaction mode between CYP1A and **ERNM**. The CYP1A1 and CYP1A2 mediated demethylation of **ERNM** both followed kinetics of substrate inhibition (Appendix A). The *K*_m_ values for demethylation catalyzed by CYP1A1 and CYP1A2 were determined as 1.0 ± 0.1 and 2.9 ± 0.5 μM, respectively (Appendix A). The *V*_max_ values for demethylation catalyzed by CYP1A1 and CYP1A2 were determined as 1.92 ± 0.04 and 3.32 ± 0.15 nmol/min/nmol P450, respectively. These results implied a relatively high affinity and catalytic efficiency for the demethylated reaction of **ERNM**. Furthermore, the similar apparent kinetic parameters for CYP1A1- and CYP1A2-mediated demethylated reaction illustrate the highly consistent catalytic behaviors of CYP1A1 and CYP1A2 toward **ERNM**. The selectivity of **ERNM** and kinetic behavior of the CYP1A-mediated methylated reaction implied that the fluorescent probe **ERNM** enables the rapid and precise monitoring of the activity of CYP1A in complex bio-samples.

### 2.5. Colocalization Assay of ERNM toward ER

The systematic evaluation of **ERNM** confirmed its application in complex biosystems, subsequent work is related to the investigation of the subcellular localization of **ERNM** in living cells. Briefly, **ERNM** was co-incubated with a commercial ER-Tracker (red) in A549 cells, and then imaged under two-photon (**ERNM**) and single-photon (ER-Tracker) excitation conditions using confocal fluorescence microscope. As shown in Figure 3, the green fluorescence signal produced by **ERNM** correlated well with the signal of commercial ER-Tracker. The overlapped fluorescence indicated that **ERNM** could image CYP1A in biosystems and exclusively target the ER. 

### 2.6. Bioimaging of CYP1A in Liver Slice

As mentioned above, CYP1A is an important metabolizing enzyme expressed in the liver that mediates the oxidative metabolism of many clinical drugs and environmental toxicants. Herein, we used **ERNM** to image CYP1A in rat liver slice by two-photon microscopy. After staining with **ERNM**, the two-photon excited fluorescence signal caused by CYP1A was detected in liver slice (Figure 4A). The multidimensional tissue images indicated the usability of **ERNM** for depth imaging CYP1A (Figure 4B). The obtained different depth of images showed the expression of CYP1A. Therefore, **ERNM** displayed good capability for deep tissue penetration, which facilitates the direct detection of CYP1A in tissue.

### 2.7. Bioimaging of CYP1A under ER Stress

Research continues to validate that the functionality level of CYP1A can be modulated under multiple pathological conditions, including nonalcoholic fatty liver disease, cancer, and type II diabetes, since CYP1A is an ER-locating protein that is associated with the production of oxidative stress [6,7]. For example, CYP1A was reported to cause an accumulation of reactive oxidative species and sometimes may work as a pro-oxidant agent in its expression region. Therefore, in the present study, the regulation of CYP1A under ER injury, which was induced by ER stress inducer dithiothreitol (DTT; Oslowski and Urano, 2011), was investigated using the developed ER-targeting two-photon fluorescent probe **ERNM**. As shown in Figure 5A,B, there were significant fluctuations in the CYP1A activity imaged by **ERNM**, and the graphical quantification of average fluorescence intensity indicated that the CYP1A activity reached the peak at 2 h in DTT-induced ER stress cell model. Meanwhile, the protein level of CYP1A followed the similar trend with the activity level of CYP1A detected by **ERNM**, implying that the changes in CYP1A activity were attributed to the regulation of protein expression of CYP1A in ER stress state. Notably, the variation trend of CHOP, a marker for the production of ER stress also increased at the beginning and then declined with time. The correlation between changes in activity and protein level of CYP1A and changes in the state of ER further confirmed the close association of ER state and ER-locating CYP1A. These results indicated that **ERNM** could image CYP1A in living cells, which is useful for evaluating the changes in CYP1A activity and protein level under ER stress and some ER-related diseases.

## 3. Materials and Methods

### 3.1. Instruments and Reagents

cDNA-expressed recombinant human CYP (rhCYP) isoforms including rhCYP1A1, rhCYP1A2, rhCYP1B1, rhCYP2A6, rhCYP2B6, rhCYP2C8, rhCYP2C9, rhCYP2C18, rhCYP2C19, rhCYP2D6, rhCYP2E1, rhCYP2J2, rhCYP3A4, rhCYP3A5, rhCYP4A11, rhCYP4F2, rhCYP4F3, and rhCYP4F12 were purchased from BD Biosciences (Woburn, MA, USA). Cofactors involved in the in vitro assay, such as *β*-nicotinamide adenine dinucleotide phosphate (NADP^+^), glucose-6-phosphate dehydrogenase, and glucose-6-phosphate were purchased from Sigma-Aldrich (St. Louis, MO, USA). The chemical inhibitors for different CYP subtypes including clomethiazole, sulfaphenazole, omeprazole, and quinidine were purchased from Sigma-Aldrich (St. Louis, MO, USA); ketoconazole and montelukast were obtained from Meilunbio (Dalian, China); N,N′,N″-triethylene thiophosphoramide (TEPA) was purchased from Aladdin (Shanghai, China). The mixed gender 150-donor pooled human liver microsomes were purchased from BIOIVT elevating science (Baltimore, MD, USA). ER-Tracker Red was purchased from Yeasen Biotechnology Co., Ltd. The solvents for liquid chromatography were obtained from Tedia (Fairfield, OH, USA). All other reagents were of the highest grade as commercially available.

^1^H-NMR and ^13^C-NMR spectra were recorded using a Bruker spectrometer (in CDCl_3_). High-resolution mass spectral (HR-MS) analyses were measured using an AB sciex X500R. Incubation samples were measured on a BioTek Synergy H1 Hybrid Multi-Mode Microplate Reader. The cells and tissue samples were imaged by a FV1000 MPF confocal laser scanning microscope (Olympus, Tokyo, Japan).

### 3.2. Synthesis of Probe



4-bromo-1,8-naphthalic anhydride (138.5 mg, 0.50 mmol) and *N*-tosylethylenediamine (117.8 mg, 0.55 mmol) were dissolved in 30 mL ethanol. The reaction mixture was refluxed for 16 h, the solvent was removed in vacuum, and the residual solid was purified by chromatography (silica gel, EtOAc–hexane as eluent, 1:4, *v*/*v*) to afford 160.5 mg **ERNBr**. ^1^H NMR (600 MHz, CDCl_3_) *δ* 8.62–8.54 (m, 2H), 8.32 (d, *J* = 7.8 Hz, 1H), 8.05 (d, *J* = 7.8 Hz, 1H), 7.86 (dd, *J* = 8.4, 7.4 Hz, 1H), 7.52 (d, *J* = 8.2 Hz, 2H), 6.73 (d, *J* = 8.0 Hz, 2H), 5.11 (t, *J* = 5.3 Hz, 1H), 4.29–4.25 (m, 2H), 3.49 (dd, *J* = 11.1, 5.5 Hz, 2H), 1.95 (s, 3H). ^13^C NMR (150 MHz, CDCl_3_) *δ* 164.06, 163.99, 142.68, 137.17, 133.64, 132.28, 131.42, 131.16, 130.73, 130.58, 129.18, 128.96, 128.17, 126.67, 122.61, 121.71, 42.46, 39.23, 21.14. HRMS (ESI positive) calcd for [M + H]^+^ 473.0165, found 473.0176.

A mixture of **ERNBr** (118 mg, 0.25 mmol) and K_2_CO_3_ (347.5 g, 2.5 mmol) in 30 mL CH_3_OH was refluxed for 10 h, the solvent was removed in vacuum, and the residual solid was purified by chromatography (silica gel, EtOAc–hexane as eluent, 1:4, *v*/*v*) to afford 80.5 mg **ERNM**. ^1^H NMR (600 MHz, CDCl_3_) *δ* 8.58 (d, *J* = 8.3 Hz, 1H), 8.50 (d, *J* = 7.2 Hz, 1H), 8.47 (d, *J* = 8.3 Hz, 1H), 7.71 (t, *J* = 7.8 Hz, 1H), 7.52 (d, *J* = 8.1 Hz, 2H), 7.05 (d, *J* = 8.3 Hz, 1H), 6.71 (d, *J* = 8.0 Hz, 2H), 5.26 (t, *J* = 4.8 Hz, 1H), 4.27–4.23 (m, 2H), 4.16 (s, 3H), 3.47 (dd, *J* = 10.8, 5.2 Hz, 2H), 1.92 (s, 3H). ^13^C NMR (150 MHz, CDCl_3_) *δ* 164.92, 164.41, 161.15, 142.55, 137.01, 133.87, 131.80, 129.42, 129.14, 129.06, 126.68, 126.02, 123.44, 121.89, 114.56, 105.28, 56.36, 42.83, 38.93, 21.09. HRMS (ESI positive) calcd for [M + H]^+^
*m*/*z* 425.1166, found *m*/*z* 425.1167.

### 3.3. The pH Effects on Fluorescence Signal of Probe Substrate ERNM and Probe Product ERNO

The effect of pH variation, within the range from 3 to 12, on the fluorescence intensity of probe substrate **ERNM** and the corresponding probe product **ERNO** was examined (the in-system concentration of **ERNM** and **ERNO** in the incubation system is 10 μM). The intensity of fluorescence signal at 558 nm that was excited at 450 nm was recorded. Data were analyzed using GraphPad Prism 8.0 (GraphPad Software, Inc., San Diego, CA, USA).

### 3.4. The Influence of Analytes on the Fluorescence Response of ERNM

The effect of various analytes in reaction system including common ion (Mn^2+^, Ca^2+^, Mg^2+^, Ni^2+^, Zn^2+^, Sn^4+^, K^+^, Fe^3+^, Na^+^, Ba^2+^), amino acid (Ser, Trp, Gly, Arg, Cys), and GSH on the fluorescence response of **ERNM** was examined at the concentration of 10 µM in the presence of NADPH-generating systems. The fluorescence signal of incubation samples in the presence of various analytes and recombinant human CYP1A1 and CYP1A2 were compared with samples in the absence of analytes. The total volume of the incubation mixture is 200 μL.

### 3.5. In Vitro Assay for CYP Activity

The incubation sample consisted of NADPH-generating system (including NADP^+^, magnesium chloride, glucose-6-phosphate, and glucose-6-phosphate dehydrogenase), and human liver microsomes or recombinant human CYP enzyme in potassium phosphate buffer (100 mM, pH 7.4). The total volume of the incubation mixture is 200 μL. Fluorescent substrate was dissolved in DMSO, and was pre-diluted to the working concentrations. If necessary, blank samples without NADPH or without enzymes will be set up as negative controls. The incubation reaction was initiated by adding NADP^+^ after pre-incubation at 37 °C for 3 min. In addition, ice acetonitrile (100 µL) was added into the incubation samples for terminating the reaction progress. 

### 3.6. Fluorescence Response of ERNM toward CYP1A and Various Human CYPs

To reveal the enzyme specificity for **ERNM**, the human recombinant CYP enzymes including CYP1A1, CYP1A2, CYP1B1, CYP2A6, CYP2A13, CYP2B6, CYP2C8, CYP2C9, CYP2C18, CYP2C19, CYP2D6, CYP2E1, CYP2J2, CYP3A4, CYP3A5, CYP4A11, CYP4F2, CYP4F3, and CYP4F12 (15 nM) were incubated with **ERNM** (the in-system concentration of **ERNM** in the incubation system is 10 μM) in 200 μL of the incubation mixture in the presence of NADPH-generating systems. After incubation for 30 min at 37 °C, the dealkylated metabolite was measured by quantifying the intensity of fluorescence signal at 558 nm that was excited at 450 nm. Data were analyzed using GraphPad Prism 8.0 (GraphPad Software, Inc., San Diego, CA, USA).

To confirm the linear range of **ERNM** in response to different enzyme concentrations of CYP1A1 and CYP1A2, the CYP1A1 (0.5, 1, 2.5, 5, 7.5, 10, 15, and 20 nM) and CYP1A2 (0.5, 1, 2.5, 5, 7.5, 10, 15, and 20 nM) were incubated with **ERNM** (the in-system concentration of **ERNM** in the incubation system is 10 μM) in the presence of NADPH-generating systems, respectively. The total volume of the incubation mixture is 200 μL. After incubation for 30 min, the formation of the dealkylated metabolite for **ERNM** was measured by quantifying the intensity of fluorescence signal at 558 nm that was excited at 450 nm. Data were analyzed using GraphPad Prism 8.0 (GraphPad Software, Inc., San Diego, CA, USA).

### 3.7. Inhibition Assay Using Selective Chemical Inhibitors for Different CYP Subtypes

The velocities of methylation of **ERNM** in HLM with the absence or presence of selective inhibitors for various CYP subtypes were measured to identify the major contribution of CYP1A toward the oxidative biotransformation of **ERNM**. In brief, **ERNM** (the in-system concentration of **ERNM** in the incubation system is 10 μM) was incubated in HLM in the absence (control samples) or presence (inhibitory samples) of selective chemical inhibitors for human different CYP subtypes in potassium phosphate buffer (100 mM, pH 7.4) with NADPH-generating system. The incubation mixture is 200 μL. The involved chemical specific inhibitors and their work concentrations in the incubation system were listed as follows: α-Naphthoflavone (2.5 μM), a selective chemical inhibitor for CYP1A subfamily that consists of CYP1A1 and CYP1A2 subtypes; N,N′,N″-triethylene thiophosphoramide (TEPA, 50 μM), a selective chemical inhibitor for CYP2B6; omeprazole (20 μM), a selective chemical inhibitor for CYP2C19; montelukast (2 μM), a selective chemical inhibitor for CYP2C8; sulfaphenazolum (10 μM), a selective chemical inhibitor for CYP2C9; quinidine (10 μM), a selective chemical inhibitor for CYP2D6; clomethiazole (50 μM), a selective chemical inhibitor for CYP2E1; ketoconazole (1 μM), a selective chemical inhibitor for CYP3A subfamily that mainly consists of CYP3A4 and CYP3A5 subtypes. Inhibition by TEPA for CYP2B6 were examined by adding **ERNM** after pre-incubation with NADPH-generating system at 37 °C for 30 min. The intensity of fluorescence signal at 558 nm for samples treated with the absence or presence of selective inhibitors for various CYP subtypes was measured. In addition, the residual activity of samples was calculated by the percentage of fluorescence signal for the chemical inhibition group to the control group, with the absence of specific inhibitors. Data were analyzed using GraphPad Prism 8.0 (GraphPad Software, Inc., San Diego, CA, USA).

### 3.8. Enzyme Kinetic Analysis

Briefly, CYP1A1 (5 nM) or CYP1A2 (5 nM) were incubated with **ERNM** (1, 5, 10, 15, 50, 100, 200 μM) in 200 μL of the incubation mixture in the presence of NADPH-generating systems. After incubation for 30 min, the dealkylated metabolite was measured by quantifying the intensity of fluorescence signal at 558 nm that was excited at 450 nm. Data were analyzed using GraphPad Prism 8.0 (GraphPad Software, Inc., San Diego, CA, USA).

### 3.9. Cell Culture and Imaging

A549 cells were grown in RPMI-1640 culture medium (containing 10% FBS). A549 cells were seeded and incubated at 37 °C overnight. Then, A549 cells divided into different experimental groups were treated with DTT (dithiothreitol, the in-system concentration of DTT in the incubation system is 2.5 mM) for 0, 1, 2, and 4 h. The adherent cells were washed, and **ERNM** was added into the cell culture media (the in-system concentration of **ERNM** in the incubation system is 25 μM) and incubated for 60 min. Then, the cells were rinsed with PBS for at least 3 times to remove the extracellular probe. The cell images were obtained using confocal microscope (Olympus, FV1000), and the excitation wavelength was set at 800 nm and fluorescent emission windows of 520–560 nm. 

### 3.10. Preparation of Liver Slices and Imaging

Slices were prepared from the liver of 8-week-old male SD rat. The liver was cut into squares of appropriate size, and an appropriate amount of OCT embedding agent was added for immersion in the tissue. Then, the tissue was frozen by nitrogen for 10–20 s. Thereafter, the tissue ice cubes were removed and immediately placed in the constant cold box microtome for the frozen section. The liver tissues were cut into 100 µm in thickness. The liver slices were incubated with **ERNM** or only incubated with PBS (the in-system concentration of **ERNM** in the incubation system for liver slices is 25 μM) at 37 °C for 60 min. Then, the slices were rinsed with PBS for at least 3 times to remove the extracellular probe. Images for liver tissues were obtained using confocal microscope (Olympus, FV1000), and excitation wavelength was set at 800 nm and fluorescent emission windows of 520–560 nm. 

## 4. Conclusions

In summary, we developed a selective two-photon fluorescent molecular tool **ERNM** for the visualization of ER-targeting CYP1A. **ERNM** could target the ER and monitor the fluctuation of CYP1A activity in living cells and tissues. The detection ability of **ERNM** for the functionality level of CYP1A under ER stress was further confirmed. Based on the ER-targeting two-photon probe for CYP1A, the close association of ER state and ER-locating CYP1A was confirmed, which would promote the understanding of CYP1A biofunctions in ER-related diseases.

## Data Availability

Not applicable.

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
