# Peer review of "Endoplasmic Reticulum-Targeting Two-Photon Fluorescent Probe for CYP1A Activity and Its Imaging Application in Endoplasmic Reticulum Stress"

_molecules, 2023, doi:10.3390/molecules28083472_

Round 1

Reviewer 1 Report

Shi et al. designed and synthesized an ER-targeting two-photon probe for detecting CYP1A activity, and the probe was further applied for ER stress imaging in living cells and liver tissue. The probe holds the potential to be applied for understanding the biofunction of CYP1A in various ER-related disease. Overall, the paper was well-organized, and the authors also carried out sufficient experiments to support their claims. Therefore, I recommend the paper to be accepted by molecules after addressing the following minor issues. 

1.Time-dependent experiment of the probe incubated with CYP1A should be provided.

2.In cell imaging experiment,  A549 cell line was chosen. Is CYP1A overexpressed in A549 cells? Moreover,  for tissue imaging, liver slice was selected. Whether CYP1A is also overexpressed in HepG2 cells (a liver cancer cell line)? 

3.In 2.7 bioimaging experiment, DTT was utilized for setting up ER stress cell model. It is strongly suggested to cite the reference literature. Besides, the full name of DTT should be provided, and the function of DTT should be explained in detail.

4.The authors only provided two-photon imaging data. Normally, one-photon imaging data should also be provided. 

Author Response

1. Time-dependent experiment of the probe incubated with CYP1A should be provided. Reply: Time-dependent responses for the ERNM incubated with CYP1A1 (7.5 nM) and CYP1A2 (7.5 nM) were added in the supplementary data (Supplementary Figure 1). The fluorescence intensity at 558 nm increased linearly when prolonging the incubation time with CYP1A1 and CYP1A2, respectively. 2. In cell imaging experiment, A549 cell line was chosen. Is CYP1A overexpressed in A549 cells? Moreover, for tissue imaging, liver slice was selected. Whether CYP1A is also overexpressed in HepG2 cells (a liver cancer cell line)? Reply: The expression level of CYP1A in the A549 cell line and liver slice were high. In the present study, consideration that the A549 cell line was derived from different organs from liver and therefore may indicate a wide range of applications for the probe, we used A549 cell line in spite of the relatively high expression level of CYP1A in HepG2 cell line. 3. In 2.7 bioimaging experiment, DTT was utilized for setting up ER stress cell model. It is strongly suggested to cite the reference literature. Besides, the full name of DTT should be provided, and the function of DTT should be explained in detail. Reply: Thanks for your kind suggestion. The relevant contents have been added in the revised manuscript. Please see the Page 7. 4. The authors only provided two-photon imaging data. Normally, one-photon imaging data should also be provided. Reply: The one-photon imaging of CYP1A was seriously disturbed by the relative high background interference in cell and tissue. Therefore, in the present study, we only provided two-photon imaging data.

Reviewer 2 Report

In this manuscript, the authors reported a endoplasmic reticulum-targeting two-photon fluorescent probe to evaluate CYP1A activity both in living cells and tissues. The manuscript was well constructed and the experiments were well performed, but there are still some concerns need to be addressed before it can be accepted.

1.      The imaging probes and methods was not well summarized and introduced in the Introduction part, please add corresponding texts.

2.      Please specify more clearly about the targeting mechanism of the probe.

3.      Please provide the data graph of NMR and MS characterizations of the probe.

4.      Colocalization coefficient needs to be supplemented to illustrate the targeting capability of the probe.

5.      For bioimaging of CYP1A under ER stress, a standard method should be performed to compare the feasibility and accuracy of the imaging probe.

6.      Please add scale bar for Figure 4B.

Author Response

1. The imaging probes and methods was not well summarized and introduced in the Introduction part, please add corresponding texts. Reply: The imaging probes for CYP1A and its relative introduce have added in the introduction part. Please see Page 2. 2. Please specify more clearly about the targeting mechanism of the probe. Reply: The methyl sulphonamide moiety was introduced into ERNM to assist the probe in accumulating in the ER by binding to the sulphonamide receptor [Chem. Sci., 2017,8, 7025-7030; The molecular probes handbook, Life Technologies Corporation, Carlsbad, 2010]. 3. Please provide the data graph of NMR and MS characterizations of the probe. Reply: The relative data was shown in Supplementary Figure 6 to 10. 4. Colocalization coefficient needs to be supplemented to illustrate the targeting capability of the probe. Reply: The Pearson's correlation was determined as 0.7. The low correlation coefficient is mainly due to the signal deviation between two-photon (our fluorescent probe) and one-photon signals (ER-tracker), for the focal planes of single-photon imaging and two-photon imaging are not exactly the same. 5. For bioimaging of CYP1A under ER stress, a standard method should be performed to compare the feasibility and accuracy of the imaging probe. Reply: Currently, a variety of fluorescent probes for CYP1A and ER localization probes have been reported. However, the ER-targeted fluorescent probes that could detect CYP1A activity are still lacking. This is also the starting point for the construction of ER localization of CYP1A fluorescent probes. Therefore, a standard method to compare the feasibility and accuracy of the imaging probe could not performed now. However, in order to verify the feasibility and accuracy of our probe, we perform the western blot assay for the detection of CYP1A1/1A2 protein in cells treated with DTT. The change trend of protein bands at different treated group was consistent with the change trend of fluorescence signal, indicating the accuracy of our probe for CYP1A imaging. Meanwhile, the changes of ER-stress marker CHOP over time implied that there was also a certain correlation between the stress response of ER and the change of CYP1A protein level. The reliability of the probe is verified by the relevant data from different aspects. 6. Please add scale bar for Figure 4B. Reply: Thanks for your kind suggestion. The scale bar had been added.

Round 2

Reviewer 2 Report

The authors have fully addressed my concerns, therefore, I recommend its publication.

Author Response

Thank you.